# FEDERATED LEARNING WITH HETEROGENEOUS ARCHITECTURES USING GRAPH HYPERNETWORKS

## ABSTRACT

Standard Federated Learning (FL) techniques are limited to clients with identical network architectures. As a result, inter-organizational collaboration is severely restricted when both data privacy and architectural proprietary are required. In this work, we propose a new FL framework that removes this limitation by adopting a graph hypernetwork as a shared knowledge aggregator. A property of the graph hyper network is that it can adapt to various computational graphs, thereby allowing meaningful parameter sharing across models. Unlike existing solutions, our framework makes no use of external data and does not require clients to disclose their model architecture. Compared with distillation-based and non-graph hypernetwork baselines, our method performs notably better on standard benchmarks. We additionally show encouraging generalization performance to unseen architectures.

## 1 INTRODUCTION

Federated learning (FL) (McMahan et al., 2017a; Yang et al., 2019; Konečný et al., 2015; 2017) allows multiple clients to collaboratively train a strong model that benefits from their individual data, without having to share that data. In particular, aggregating the parameters of locally trained models alleviates the need to share raw data, thereby preserving privacy to some extent and reducing the volume of data transferred. FL allows safe and efficient learning from edge nodes like smartphones, self-driving cars, and medical systems.

While successful, FL approaches have one key limitation: all clients need to share the same network architecture. As a result, they are not applicable to many important cases that require learning with heterogeneous architectures. For example, clients often run different networks on their edge devices, due to computational limitations or OS versions. Also, in some cases, clients want to keep their network architectures private, for instance, when different organizations wish to benefit from each other's access to data without having to share their proprietary architectures. In all these cases, one is interested in *federated learning with heterogeneous architectures* (FLHA).

Unfortunately, current FL approaches do not support mixtures of different architectures, but require that clients share the same model architecture. As an illustrative example, consider Federated Averaging (FedAvg), perhaps the most widely used FL technique, where model parameters of different clients are averaged on a shared server McMahan et al. (2017a). Unless all clients share the same number of parameters and layer structure, it is not clear how to average their weights. The problem of aggregating weights across different architecture is not specific to FedAvg, but also exists with other FL approaches. This limitation raises the fundamental research question: **Can federated learning handle heterogeneous architectures? And can it be achieved with clients keeping their architectures undisclosed?**

A possible alternative to FLHA could be achieved through knowledge distillation. For example, Lin et al. (2020) suggested distilling knowledge from ensembles of client models on a server using unlabeled or synthetic datasets. Although this solution could be beneficial in certain setups, it has two major limitations: (1) Client architectures must be disclosed and (2) external data must be provided. We could also design a distillation-based approach to FLHA that does not require clients to share their architectures. First, train a shared model using a standard FL approach. Once trained, distill knowledge from the shared model to train private client models using each client local data and architecture. While this solution can address the two limitations discussed above, it also has two drawbacks: (1) Training on a large shared architecture may not be feasible for clients with

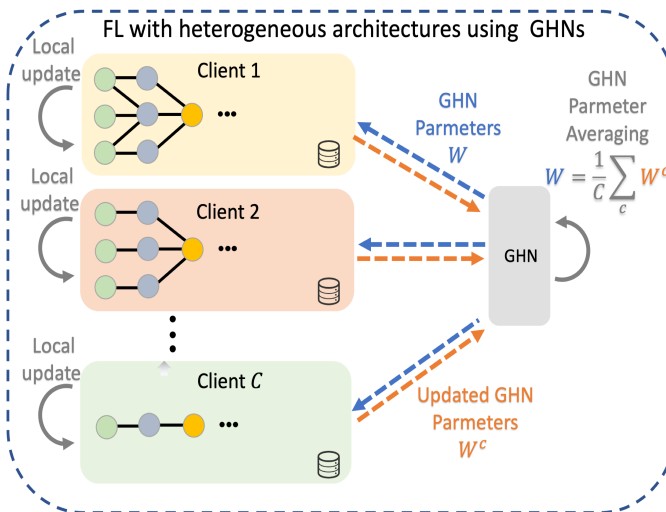

Figure 1: An overview of our approach. We tackle a federated learning setup with clients that have different architectures by using a shared Graph Hypernetwork (GHN) that can adapt to any network architecture. At each communication round, a local copy of the GHN is trained by each client using their own data and network architecture (illustrated as graphs within each client; nodes represent layers and edges represent computational flow), before returning to the server for aggregation.

limited computing power; (2) Local distillation using only local data may overfit and lose the benefits of FL. Finally, and perhaps most importantly, this approach bypasses, but does not addresses, the fundamental question: How can we learn to use knowledge about architectures when transferring information about model parameters across architectures.

In this work, we propose a general approach to FLHA based on a hypernetwork (HN) (Ha et al., 2016) that acts as the knowledge aggregator. The hypernetwork takes as input a description of a client architecture and predicts weights to populate that architecture. Unlike FedAvg, which aggregates weights in a predefined way, the hypernetwork implicitly learns an aggregation operator from data and can therefore be significantly more powerful. For providing an architecture descriptor to the HN, we propose to model client architectures as graphs where nodes represent both parametric and non-parametric layers (e.g., a convolutional layer or a summation operation) and directional edges represent network connectivity such as skip connections.

To allow the hypernetwork to process any graph, regardless of its size or topology, we use a *graph hypernetwork* (GHN) similar to Zhang et al. (2020). We therefore name our approach *FLHA-GHN*. The GHN operates on a graph representation of the architecture and predicts layer parameters at each node that represents a parametric layer. At each communication round, clients train local copies of the hypernetwork weights using their own architectures and data (Fig. 1). The updated weights are sent by the clients to the server where they are aggregated. Since different architectures have different layer compositions, representing layers as nodes allows meaningful knowledge aggregation across architectures. This forms an improved hypernetwork model that uses knowledge gleaned from different network types and datasets, to populate them with improved parameters. Critically, client architectures are not communicated.

The ability to generalize across different client architectures is crucial for successful FLHA. Recent results in theory of Graph Neural Networks (GNNs) Yehudai et al. (2021) suggest that GNNs can indeed generalize across input distributions. We discuss this issue and demonstrate experimentally that our approach generalizes to unseen architectures in certain cases – allowing clients to modify their architectures after federation has occurred, without the need for client-wide FL retraining.

In an extensive experimental study on three image datasets, we show that applying FLHA-GHN to FL with heterogeneous-architecture clients outperforms a distillation-based baseline, and a non-GNN based HN architecture (Shamsian et al., 2021), by a large margin, and that margin generally grows when the amount of local data decreases. We further show that FLHA-GHN provides a large benefit in generalization to unseen architectures, improving the accuracy of converged models and drastically shortening the time needed for convergence. Such generalization could allow new clients, with new architectures, to benefit from models trained on different data and different architectures, lowering the bar for deploying new, personalized, architectures.

## 2 PREVIOUS WORK

**Federated Learning.** Federated learning McMahan et al. (2017a); Kairouz et al. (2019); Yang et al. (2019); Mothukuri et al. (2021) is a learning setup in which multiple clients collaboratively train individual models while trying to benefit from the data of all the clients without sharing their data. The most well known FL technique is federated averaging, where all clients use the same architecture, which is trained privately and then sent to the server and averaged with other locally trained models. Many recent works have focused on improving privacy McMahan et al. (2017b); Agarwal et al. (2018); Li et al. (2019); Zhu et al. (2020) and communication efficiency Chen et al. (2021); Agarwal et al. (2018); Chen et al. (2021); Dai et al. (2019); Stich (2018). Another widely studied setup is the heterogeneous client data setup Hanzely & Richtárik (2020); Zhao et al. (2018); Sahu et al. (2018); Karimireddy et al. (2020); Zhang et al. (2021). To solve this problem, personalized FL (pFL) methods were proposed that adapt global models to specific clients Kulkarni et al. (2020) . Most related to the current work is the recent work of Shamsian et al. (2021) that also used hyper-networks. Their method considers a simple case of clients with varying architectures, where three pre-defined simple architectures are "hard" encoded into the structure of the hypernetwork. In contrast, our framework allows clients to use a variety of layers and computational graphs, and facilitates better weight sharing as the same layers are used in different architectures. In the experimental section 5 we show that our approach outperforms Shamsian et al. (2021) by large margins.

**Hypernetworks.** HyperNetworks (HN) Klein et al. (2015); Ha et al. (2016) are neural networks that predict input conditioned weights for another neural network that performs the task of interest. HNs are widely used in many learning tasks such as generation of 3D content Littwin & Wolf (2019); Sitzmann et al. (2019), neural architecture search Brock et al. (2017) and language modelling Suarez (2017). More relevant to our work are Graph Hypernetworks (GHNs) - hypernetworks that take graphs as an input. Nachmani & Wolf (2020) used GHNs for molecule property prediction. Even more relevant is the use of GHNs for Neural Architecture Search (NAS) Zhang et al. (2020). In our work, we adapt GHNs to FLHA with unique challenges arising from the problem setup.

## 3 PROBLEM DEFINITION

Traditional FL addresses the setup of $C$ clients working together to improve their local models, while each client $c \in \{1, \ldots, C\}$ has access only to its local data samples $\{(x_{cj}, y_{cj})\}_{j=1}^{n_c}$, sampled from client specific data distributions $\mathcal{P}_c$, $c = 1, \ldots, C$.

In this paper, we generalize FL to FL with Heterogeneous Architectures (FLHA). In this setup, clients can use different network architectures $f_1(\cdot; \theta_1), f_2(\cdot; \theta_2), \ldots, f_C(\cdot; \theta_C)$. Here, $f_c(\cdot; \theta_c)$ is a neural architecture from some predefined family of models with parameters $\theta_c \in \mathbb{R}^{m_c}$ (see further discussion on architecture families in Section 4) . Moreover, we assume that all clients are connected to a server, and that the clients can share information among themselves only through the server. Importantly, in FLHA we are interested in the setup where both the architectures and the data cannot be transferred due to privacy concerns.

Our goal is to solve the following minimization problem:

$$(\theta_1^*, ..., \theta_C^*) = \text{argmin}_{(\theta_1, ..., \theta_C)} \sum_{c=1}^{C} \mathbb{E}_{(x,y) \sim \mathcal{P}_c} \ell(f_c(x; \theta_c), y), \qquad (1)$$

for a suitable loss function $\ell(y, y')$.

## 4 APPROACH

### 4.1 OVERVIEW

Standard FL methods rely on aggregating model parameters and are not directly applicable when clients use different architectures. In our FLHA setup, the parameter vectors $\theta_c$ of different clients have different shapes and sizes, and as a result, a direct aggregation can be meaningless or not well defined. To address this issue, we re-parameterize the weights $\theta_c$ of a model $c$ as the output of a hypernetwork. The hypernetwork serves as a trainable knowledge aggregator. The parameters of this

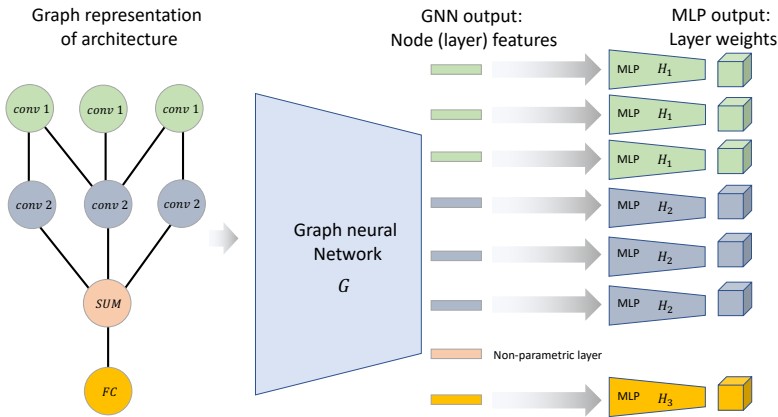

Figure 2: The GHN architecture. GHN is composed of two sub-networks: a Graph neural network (GNN) $G$ and a set of MLPs $\{H_l\}_{l \in L}$, one for each layer type. We input a graph representation of an architecture to the GHN, with the colored node features corresponding to different layer types. We then apply the GNN to produce node features. Finally, the node features for the parametric layers are further processed by an MLP $H_l$ to attain the final layer weights.

operator are learned from data using updates from all clients. Since the output of the hypernetwork must fit a given architecture $f_c$, the hypernetwork should take as input a representation of $f_c$. Here, we advocate graphs as a natural representation for neural architectures (in agreement with Zhang et al. (2020)) and apply graph hypernetworks to process them.

The workflow of training our FLHA-GHN is illustrated in Figure 1. At each communication round, several steps are followed. (1) First, a server shares the current weights of the GHN, $W$, with all clients; (2) Each client $c$ uses the GHN to predict weights $\theta_c$ for its own specific architecture $f_c$, and updates the GHN weights $W_c$ locally using its own architecture and data; (3) Each client sends the locally updated GHN weights to the server; (4) Weight averaging is performed on the server.

## 4.2 METHOD

**Representing neural architectures as graphs.** A neural architecture can be represented in many different ways. Early chain-structured architectures such as VGG Simonyan & Zisserman (2014) and AlexNet Krizhevsky et al. (2012) can easily be represented as sequences of layers. Recently, several architectures that have richer connectivity structures have been proposed Ronneberger et al. (2015); Targ et al. (2016); Huang et al. (2017). Sequential representation is not suitable for these architectures. In contrast, all neural architectures are computational graphs, which makes a graph representation a much more natural choice. We use graph representations to ensure generality in architecture space.

Given a neural architecture, we represent it as a graph $\mathcal{A} = (\mathcal{V}, \mathcal{E}, \mathcal{X})$ in the following way. The set of vertices $\mathcal{V}$ contains a vertex $v$ for each parametric layer in the architecture. For example, a convolutional layer with weights of dimensions $3 \times 3 \times 64 \times 128$. In a similar way, we use another type of nodes to represent non-parametric operation in the network, such as summation or concatenation of matrices, which are frequently used in ResNet-like architectures. See example in Figure 2. The set of edges $\mathcal{E}$ represents the computational flow of the architecture: there is an edge between the nodes $v$ and $u$ (i.e., $e = (v, u) \in \mathcal{E}$) if the output of the layer represented by $v$ is the input of a layer represented by $u$. $\mathcal{X} \in \mathbb{R}^{|\mathcal{V}| \times k}$ is a matrix that holds the input node features. Initially, each node is equipped with categorical (one-hot) features indicating the layer type they represent, denoted by $L = \{l_1, \dots, l_k\}$. Each categorical layer type specifies the following: linear/conv, stride, kernel size, activation and feature dimensions. As an example, the convolution layer discussed above may be represented as $l_1$. According to the desired architecture family, the nodes in the graph can represent different computational blocks of different granularity. Granularity can range from a single layer to complex blocks, so nodes can even represent complex mechanisms like attention. Throughout this paper, we consider the ResNets architecture family and model layers as nodes.

**Architecture.** Originally introduced for neural architecture search, a GHN Zhang et al. (2020) is a generalization of standard hypernetworks that allows generating weights for heterogeneous network architectures. The GHN weight prediction can be broken down into two stages: (1) Processing a graph representation of an architecture using a GNN and then (2) Predicting weights for each layer in that architecture. We now explain these two stages in detail.

In the first stage, our hypernetwork takes as input a graph representation $\mathcal{A}$ of an architecture and processes it using a $T$-layer graph neural network $G(\cdot; W_G)$ with learnable parameters $W_G$ (we use $T = 6$ in our implementation). This process outputs latent representations $h_v^T$ for each node $\{v \in \mathcal{V}\}$. We use maximally expressive message passing GNN layers, as introduced in Morris et al. (2019). These layers have the following form:

$$h_v^{(t)} = \sigma \left( A^{(t)} h_v^{(t-1)} + B^{(t)} \sum_{\{u|(u,v)\in\mathcal{E}\}} h_u^{(t-1)} + b^{(t)} \right). \tag{2}$$

Here, $t \in \{1, \ldots, T\}$ represents the depth of the layer, $A^{(t)}, B^{(t)}, b^{(t)}$ are layer-specific parameters. and $\sigma$ is a non-linear activation function such as ReLU. As mentioned above, we denote the concatenation of the parameters $A^{(t)}, B^{(t)}, b^{(t)}, t \in \{1, ..., T\}$ as $W_G$. As shown in several recent works, the node features extracted by such GNNs are a representation of the local neighbourhood around each node (Xu et al., 2018; Morris et al., 2019; Yehudai et al., 2021).

At the second stage, we use a set of MLPs $\{H_l(\cdot; W_H^l)\}_{l \in L}$ with learnable parameters $W_H^l$, $l \in L$ to map latent node representations $h_v^T$ to layer weights $\theta^v$:

$$\theta^v = H_{l(v)}(h_v^T; W_H^{l(v)}), \quad v \in \mathcal{V}, \tag{3}$$

where $l(v)$ is a categorical variable that determines the type of the layer represented by the node $v$. We note that a single MLP will not suffice since there are multiple layer types with different output sizes. We denote the concatenation of $\{W_H^l\}_{l \in L}$ as $W_H$. For a particular client $c$, the weights $\{\theta^v\}_{v \in \mathcal{V}}$ are concatenated to form the client's weight vector $\theta_c(\mathcal{A}_c; W_G, W_H)$ mentioned above.

**Objective and training procedure.** Based on the GHN formulation, we can now state our training objective: we are looking for optimal GHN paramters $(W_G^*, W_H^*)$ that simultaneously minimize the empirical risk of all clients:

$$(W_G^*, W_H^*) = \text{argmin}_{(W_G, W_H)} \sum_{c=1}^{C} \sum_{j=1}^{n_c} \ell(f_c(x_{cj}; \theta_c(\mathcal{A}_c; W_G, W_H)), y_{cj}). \tag{4}$$

The training procedure of our GHN is based on local updates of the GHN weights, performed by all clients, followed by a GHN weight averaging process on the shared server. More specifically, the local optimization at each client $c$ aims to solve the following client-specific minimization problem:

$$(W_G^{c*}, W_H^{c*}) = \text{argmin}_{(W_G, W_H)} \sum_{j=1}^{n_c} \ell(f_c(x_{cj}; \theta_c(A_c; W_G, W_H)), y_{cj}), \tag{5}$$

and performs a predefined number of SGD iterations (or similar gradient-based optimization method). The locally updated weights are then averaged at the server side and redistributed to the clients for further updates. The procedure is summarized in Algorithm 1.

We note that FedAvg can be seen as a specific instance of FLHA-GHN. To see that, consider the standard FL setting where all client architectures are the same and the GNN implements the identity map. Consequently, all clients will have the same latent node representations $h_v^0$: the 1-hot encoding of the layer. If the MLPs $\{H_l(\cdot; W_H^l)\}_{l \in L}$ are all implemented as a linear mapping (i.e., $\theta^v = H_{l(v)}(h_v) = W_{l(v)}h_v$) then averaging their parameters $W_{l(v)}$ is equivalent to directly averaging the client network parameters.

**Hypernetwork weight initialization.** Poor weight initialization of CNNs can have a detrimental effect to the training behavior causing either vanishing or exploding gradients. This has been carefully studied in Glorot & Bengio (2010); He et al. (2015b). When the network weights are outputs of

---

**Algorithm 1:** FLHA-GHN

---

**Input:** R: number of communication rounds, C: number of clients, L: local updates.
Initialize shared GHN weights ;
**for** *r = 1,...,R* **do**
    Server shares current GHN weights $(W_G, W_H)$ with all clients $c \in \{1, ..., C\}$ ;
    **for** *c = 1,...,C* **do**
        Update GHN weights by local optimization for L update steps on client $c$ (see Eq. equation 5);
        Send updated GHN weights $(W_G^c, W_H^c)$ to the server ;
    **end**
    Average GHN weights: $W_G \leftarrow \frac{1}{C} \sum W_G^c, W_H \leftarrow \frac{1}{C} \sum W_H^c$ ;
**end**

---

another network, proper initialization of the hypernetwork parameters must be carried out so as to output the proper main network initialization. This has been observed by Chang et al. (2019) who proposed a variance formula. We empirically found that initializing $W_H$ using zero bias and Xavier-normal initialization multiplied by $\sqrt{2c_{\text{in}}/d_{\text{lat}}}$ translates to an accurate Kaiming initialization of the client network. Here $c_{\text{in}}$ and $d_{\text{lat}}$ are the number of channels in each layer and the hidden layer dimension of $H_l$. See supplementary for more details.

**Inference and local refinement.** At inference time, the trained GHN is applied to the graph representation of a given architecture $\mathcal{A}$, yielding architecture-specific weights $\theta_c(\mathcal{A}; W_G, W_H)$. In contrast to the training stage, where these weights are used as-is, at the inference stage, they can be used as an initialization and then locally refined by using the client's local dataset. The amount of improvement depends on the amount of local data. Furthermore, since this data has already been used for training the shared GHN model, minor improvements should be expected before overfitting might occur. We found that a single refinement epoch of just the linear prediction head worked well.

### 4.3 IMPLEMENTATION DETAILS

For full reproducibility of our results, we will release code upon publication. We ran an extensive hyperparameter search using Biewald (2020) for a 4 architecture setup using a fixed number of 500 epochs. We found the optimal values (which were used in all our experiments) to be: the GNN introduced in (Morris et al., 2019) with $T = 6$ layers, a latent dimension of $51$, SGD with a learning rate of $0.009$ and cosine scheduling. Inspired by Zhang et al. (2020), we experimented with both synchronous and asynchronous message-passing that respects the directional graph structure, yet experiment did not indicate an advantage to the latter. For the client architectures, we define $k = 10$ different layer types. The different architectures and non-parametric layers are described in detail in the supplementary material. For each layer type, a 2-layer MLP hypernetwork is used with a latent dimension of 16, and a leaky-ReLU activation function. We adopted the idea from Littwin & Wolf (2019) and separately predict the scale $s$ and weight $W$ for each node so that the layer parameters become $sW$. Implementation was done in PyTorch Paszke et al. (2017) using PyTorch Geometric (Fey & Lenssen, 2019) and training was done on a cluster with NVIDIA DGX V100 GPUs. The range of parameter weep and implementation details for baselines are provided in the supplementary.

## 5 EXPERIMENTS

Here we describe a comparative experimental study of our approach. In Section (5.1) we review the methods that we compare with. Sections 5.2 & 5.3 provide results for two applications. Finally, we show ablations including generalization to unseen architectures (Sec. 5.4), the effect of communication rate (Sec. 5.5), and the importance of the GNN (Sec. 5.6). In the supplementary material, we also demonstrate strong results with our method in unbalanced data distribution scenarios.

We demonstrate our FLHA performance on two tasks. For natural image classification we use two standard benchmarks in FL: CIFAR-10 and CIFAR-100. Another task involves disease classification in chest x-ray scans. Each experiment consists of four different network architectures, named Arch 0-3. Arch 0 is a standard ResNet18 He et al. (2015a) and the other 3 are variants of it: Arch 1 has 10 layers and no skip connections, Arch 2 has 12 layers and skip connections are used only in the *first* 8

layers, and Arch 3 has 12 (different) layers with skip connection in the *last* 8 layers. Detailed designs can be found in Figure 5 in the supplementary. In all experiments, we split the data (either uniformly or following a Dirichlet distributions, as indicated in the setup) between 4, 8, and 16 clients while maintaining an even distribution of the different architectures among clients (for instance, in the 8 clients case, we use 2 clients for each architecture).

## 5.1 METHODS IN COMPARISON

Since we are the first to address the FLHA setup (FL with heterogeneous architectures without disclosing architectures or relying on external data), we compare our performance with two variants of previous methods that we adapted to the FLHA setup. (1) A local-distillation baseline, which we name **FLHA-distiliation**. As a variant of the distillation technique proposed by Lin et al. (2020), this baseline is based on locally available data and does not require the clients to disclose their architectures. In detail, for the first step we use standard federated averaging McMahan et al. (2017b) to train a shared architecture. The trained model is then sent to the client for distillation using only local data. (2) A new variant of **pFedHN** (Shamsian et al., 2021), where the individual client heads where changed to account for different architectures (different number of outputs according to the number of parameters). In addition to these baselines, we also report results of **Local training** where each client performs standard (non-FL) training on its local train samples. This can be seen as a lower bound. For completion, we also include an **Upper bound** score. This is the result if all the clients used the most expressive Arch 0, and trained in a standard FL.

## 5.2 RESULTS ON CIFAR

We experiment with the widely known CIFAR-10/CIFAR-100 Krizhevsky et al. (2009) image classification datasets that contain $60,000$ $32 \times 32$ natural color images in 10/100 classes with predefined train/test split. The results under FLHA are summarized in Table 1. Our method outperforms the local distillation and pFedHN baselines consistently expect on CIFAR-10 when local data is $25\%$ where we perform similarly or slightly worse than local distillation. Importantly, as can be expected, as the amount of locally available data decreases, local-distillation deteriorates rapidly. By contrast, our proposed method shows gradual degradation, resulting in an improvement of between 17 and 25 points in accuracy over the baseline when local data percentage is at $6.25\%$. Evidently, despite being tuned to its best performing parameters per setup (dataset+split) pFedHN performs poorly. This can be attributed to weight sharing only occurring at the shared MLP responsible for the encoding the architecture, so the separate predictions heads do not benefit from federation.

## 5.3 RESULTS ON CHEST X-RAY

As discussed in the introduction, a main motivation for FLHA is to allow cross-organizational collaborations. This is especially important when multiple entities have access to valuable data that cannot be shared, but use different neural architectures for processing it. Such is the case for medical clinics and hospitals. To showcase the application of our method to such a real-world problem, we tested it using medical data of X-ray images. The Chest X-ray Wang et al. (2017) dataset comprises 112,120 frontal-view X-ray images of 30,805 unique patients with fourteen disease image labels (images can have multiple labels), extracted from medical reports using natural language processing. Table 3 reports the average AUC score across 14 binary classification tasks for our method as well as the baselines. Our method achieves consistently superior performance.

## 5.4 GENERALIZATION TO UNSEEN ARCHITECTURES

**Generalization to unseen architectures.** In standard FL, a client that did not participate in the training procedure may still benefit from the pre-trained network. We want FLHA-GHN to enjoy a similar generalization, namely, that the GHN could generalize to clients whose architecture has not been observed during training. Since neural architectures often share local structures, as with the various variants ResNets, when a new architecture is composed of local structures that have previously been seen, it may be feasible to generalize to a new composition of these known components (compositional generalization). The reason is that $T$-layer message-passing GNNs essentially encode the $T$-hop local connectivity pattern around each node. When the same $T$-hop neighborhood appears in a new architecture, possibly in a different position in the computational graph, the GNN will be able to predict reasonable weights since it was trained on such local structures (Yehudai et al., 2021).

| data % | method | CIFAR-100 | | | | CIFAR-10 | | | |
|---|---|---|---|---|---|---|---|---|---|
| | | Arch 0 Original | Arch 1 No Skip | Arch 2 Skip first | Arch 3 Skip last | Arch 0 Original | Arch 1 No skip | Arch 2 Skip first | Arch 3 Skip last |
| 25% | Upper-bound | 73.2 | | | | 93.6 | | | |
| | Local training | 49.5 | 50.2 | **52.2** | 48.2 | 86.3 | 84.6 | 84.7 | 85.4 |
| | pFedHN | 38.0 | 38.7 | 46.5 | 34.7 | 85.8 | 84.1 | 85.8 | 84.4 |
| | FLHA Distillation | 51.1 | 44.8 | 49.0 | 46.8 | 90.0 | **89.3** | **89.4** | **90.1** |
| | without graph | 55.0 | 52.1 | 45.2 | 54.8 | 88.9 | 87.5 | 87.5 | 89.2 |
| | FLHA-GHN (ours) | **56.0** | **52.5** | 50.8 | **54.4** | **90.4** | 89.0 | 87.3 | 88.5 |
| 12.50% | Upper-bound | 71.1 | | | | 93.2 | | | |
| | Local training | 36.4 | 32.5 | 38.1 | 33.5 | 79.2 | 75.6 | 74.5 | 80.0 |
| | pFedHN | 17.8 | 25.0 | 24.0 | 20.2 | 77.6 | 76.0 | 78.3 | 74.8 |
| | FLHA Distillation | 44.2 | 33.2 | 41.3 | 35.2 | 85.7 | 84.6 | 84.2 | 83.3 |
| | without graph | 55.4 | 52.9 | 49.5 | 54.3 | 89.0 | 88.2 | 86.2 | 89.1 |
| | FLHA-GHN (ours) | **52.2** | **50.3** | **49.1** | **51.9** | **89.5** | **88.1** | **86.5** | **88.5** |
| 6.25% | Upper-bound | 71.3 | | | | 92.9 | | | |
| | Local training | 22.7 | 19.4 | 22.0 | 22.0 | 76.9 | 75.6 | 75.6 | 74.5 |
| | pFedHN | 13.6 | 14.4 | 16.4 | 12.6 | 58.2 | 56.6 | 57.3 | 50.7 |
| | FLHA Distillation | 25.3 | 23.1 | 30.0 | 23.6 | 77.4 | 75.9 | 75.7 | 74.8 |
| | without graph | 49.7 | 51.6 | 34.4 | 50.6 | 87.9 | 86.1 | 83.8 | 87.5 |
| | FLHA-GHN (ours) | **50.3** | **50.4** | **46.9** | **49.2** | **86.8** | **85.6** | **83.9** | **85.4** |

Table 1: Accuracy on CIFAR-10/100 datasets

A clear benefit of such generalization is that given a new architecture, our GHN can immediately populate it to give a better initialization. To test this we ran a "leave one out" experiment using CIFAR-10 dataset, where we let 3 clients with 3 different architectures train in an FLHA fashion. We then introduce a 4th architecture and refine it using only local data. We compare against training that architecture from scratch on that same data. The results shown in Figure 3 and Table 6 in the supplementary are very encouraging. When refining on local data, the unseen architecture performance ramps up very quickly and reaches similar performance to that achieved by the architectures

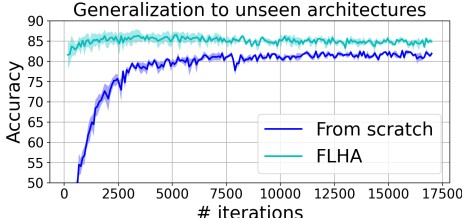

Figure 3: Generalization to unseen architectures: our method (light blue) quickly ramps up to high performance and maintains a considerable gap compared to training from scratch (dark blue).

that participated in the FLHA-GHN process. In contrast, the training-from scratch alternative takes a much longer to converge and reaches a consistently lower final performance. We further stress tested the generalization capabilities, by introducing a much smaller CNN with only 4 layers. A practical scenario of this sort is training a model on an edge device together with stronger models on the server. Table 7 in the supplementary shows the performance of the 4 architectures used in our main experiments, and how they are influenced when each is replaced by a smaller architecture. The average drop in performance of $2.2 \pm 1.4$ keeps the performance well above the local-training alternative. Despite the different architecture, when initialized with a model trained on the four main architectures, the fast convergence property persist (see Figure 7).

### 5.5 COMMUNICATION RATE

In FL, the amount of data transfer could become a bottleneck. A common remedy is to reduce the frequency of clients-server communication To study the effect of communication rates on the FLHA-GHN performance, we trained 4 architectures on CIFAR-10. At each experiment, clients train locally for $L$ epochs before sending local GHN weights to the server for averaging. Client-server communication occurs $R$ times, keeping $L \times R$ at a constant value of 1000. Figure 4 shows the results for R between 1 and

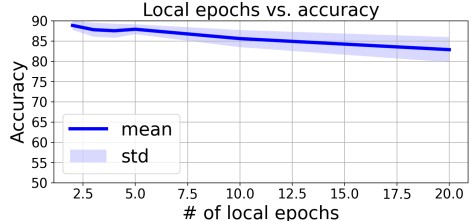

Figure 4: FLHA-GHN on CIFAR-10 with different communication rates.

20. While more frequent communication leads to better performance, we observe that even with 5 times less frequent communication the overall performance decreases by less than $1\%$.

## 5.6 GNN IMPORTANCE

For our FLHA-GHN, all models are created from the same set of layer types. The more these layers are repeated in the computational graph, the more sharing could be done, allowing better utilization of the federation. Importantly, while same layers could be directly averaged, it is important that their location within the graph is encoded. This is done via message passing in our GNN. To verify its importance, we implemented a variant of our method that does not use a GNN and instead directly averages MLPs corresponding to the same layer types. Table 2 shows this ablation, for which we used CNNs with 4,7,10 and 13 layers all of type $3 \times 3 \times 64 \times 64$ except for the first and last. With this amount of sharing, the results highlight that the GNN is instrumental for performance gains.

| data % | GNN | CIFAR10 | | | | CIFAR100 | | | |
|---|---|---|---|---|---|---|---|---|---|
| | | 4 layers | 7 layers | 10 layers | 13 layers | 4 layers | 7 layers | 10 layers | 13 layers |
| 25% | ✗ | 55.3 | 75.5 | 75.1 | 69.9 | 27.8 | 40.0 | 36.5 | 29.4 |
| | ✓ | 63.6 | 77.4 | 77.0 | 74.4 | 26.4 | 41.1 | 40.4 | 33.5 |
| 12.50% | ✗ | 60.2 | 75.5 | 73.8 | 70.1 | 32.4 | 41.5 | 41.6 | 36.8 |
| | ✓ | 67.7 | 78.2 | 77.8 | 74.4 | 36.1 | 43.5 | 42.3 | 35.0 |

Table 2: **The importance of the GNN.** Under an FLHA setup we use 4 architectures sharing a repeated layer type. The result show significant gains due to message passing informing the layer encoding with their placement within the architecture.

| data % | method | ChestX-rayWang et al. (2017) | | | |
|---|---|---|---|---|---|
| | | Arch 0 Original | Arch 1 No Skip 2 | Arch 2 Skip first | Arch 3 Skip last |
| 25% | Upper-bound | | 78.0 | | |
| | Local training | 60.5 | 61.8 | 58.7 | 60.3 |
| | pFedHN | 62.4 | 60.5 | 63.0 | 63.7 |
| | FLHA Distillation | 67.5 | 67.8 | 65.4 | 69.8 |
| | FLHA-GHN (ours) | **72.2** | **73.2** | **70.0** | **70.5** |
| 12.50% | Upper-bound | | 78.0 | | |
| | Local training | 59.5 | 59.4 | 57.1 | 59.8 |
| | pFedHN | 62.5 | 59.9 | 58.4 | 61.0 |
| | FLHA Distillation | 66.4 | 64.0 | 62.9 | 67.6 |
| | FLHA-GHN (ours) | **74.0** | **71.9** | **70.6** | **73.5** |
| 6.25% | Upper-bound | | 77.0 | | |
| | Local training | 59.2 | 58.5 | 56.9 | 58.9 |
| | pFedHN | 60.4 | 56.3 | 58.0 | 60.0 |
| | FLHA Distillation | 65.7 | **62.8** | 61.72 | 66.5 |
| | FLHA-GHN (ours) | **69.4** | 62.4 | **64.4** | **68.0** |

Table 3: Average AUC on NIH Chest X-ray dataset.

## 6 CONCLUSION AND LIMITATIONS

In this work we have proposed a new setup for federated learning called FLHA, as well as a first solution based on GHN. The results of our experiments are promising: by modeling neural networks as graphs with layers as nodes, we have shown that GNNs can utilize recurring structures and facilitate efficient federation during learning and even generalization to new architectures. There are, however, a few limitations to our solution. Firstly, it is limited to architectures built from predefined building blocks. In theory, this set could be arbitrarily large, but architectures that use disjoint sets of nodes may lose efficiency when sharing knowledge. Secondly, as indicated by the upper-bound performance, our solution still is not yet comparable with a same-architecture FL performance. Finally, a more robust aggregation technique would alleviate the undesirable degradation of performance when local data is limited. Our hope is that this research will lead to further study of this important setup.

**Ethics Statement.** In this paper, we study a federated learning setup in which the clients use different neural architectures. We term this setup Federated Learning with Heterogeneous Architecture (FLHA) and propose a graph hypernetwork solution. By relaxing the single architecture constraint, FLHA-GHN could facilitate useful cooperation beyond a single organization and create opportunities for smaller-scale entities to train efficient models. We believe that medical applications would be the primary beneficiary of our framework. While we include experiments on image classification, our contributions are not task-specific and could be applied to a variety of applications. We expect that our method will be used for positive ends, but similarly to all federated learning approaches, adversarial behavior is possible.

**Reproducibility Statement.** To guarantee reproducibility we will release code upon publication. The main method as well as baselines described in the paper are hosted in a single repository and run from a docker container. Implementation details along with optimal parameters for the main method are described in section 4.3. A discussion on how to initialize the hypernetwork is included in 4.2 and demonstrated in C. Network architectures are described in A. Hyper-parameters search for FLHA-GHN is described in D. Implementation details for the local distillation and pFedHN baselines are given in E and F, respectively.

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

## A NETWORK ARCHITECTURES

Each experiment in this paper used four different types of architectures split among the different clients plus an additional small architecture for the stress test. There are ten different types of nodes (layers) in each architecture. Figure 5 shows the architectures used in the CIFAR-10/100 experiments. The types of convolutional layers are denoted by " c<channels_in>_<channels_ out>_<kernel_size>_<stride>". In the chest x-ray experiment, where the input images are grayscale and of size $224 \times 224$ the first convolutional layer type is "c1_64_k7_s2" type.

CIFAR-10, CIFAR-100, and Chest X-rays, respectively, use different types of linear layers with 10, 100, and 14 output dimensions.

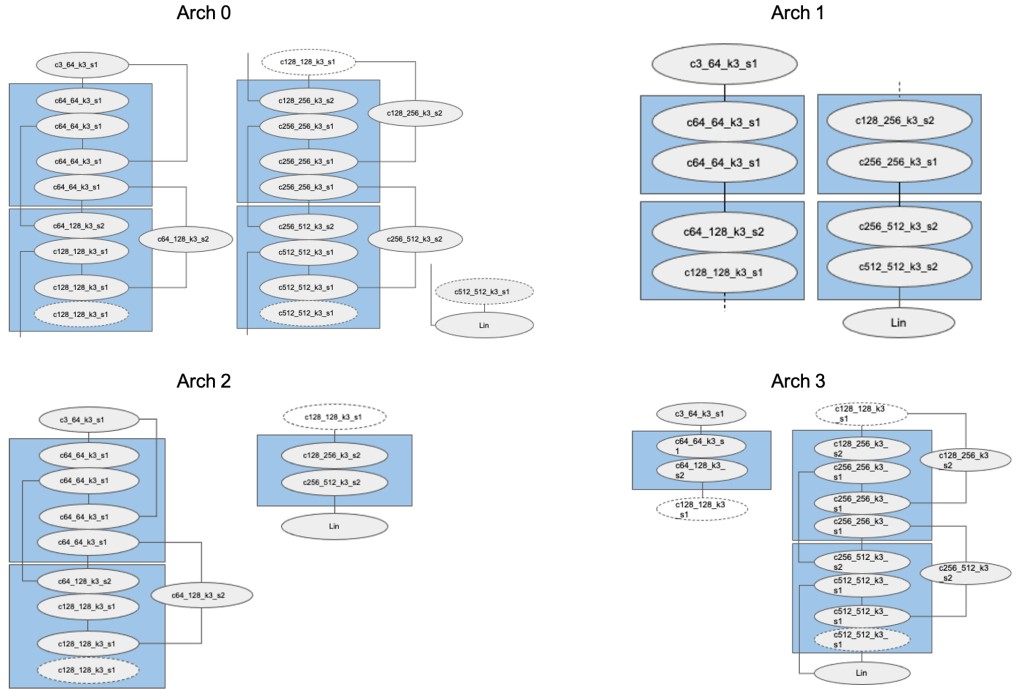

Figure 5: The four client architectures used in CIFAR-10/100 experiments.

## B NON PARAMETRIC LAYERS

Since our main architectures all use residual connections and same activation type, the graph connectivity suffices to express the architecture. However, in general, different non-parametric layers might be desired. For that, we implemented three versions of ResNet in which several different residual layers were replaced by concatenation, resulting in a hybrid addition-concatenation. To that end, we included two new non-parametric types of nodes in our layer type set: "add" and "concat". Participating in message passing, these nodes produce latent embedding, depending on where they reside in the graph. No additional parameters are required for them. We trained these 3 architectures together with the vanilla ResNet with four and eight clients with the CIFAR10 and CIFAR100 datasets. On CIFAR10, the average results by architecture type are: 88.96, 89, 89, 89.2 for 4 clients and 88.6, 88.1, 88.3, 86.8 for 8 clients. On CIFAR100, the results are: 56, 56.2, 56.5, 48.200 for 4 clients and 50.6, 50.4, 47.7, 46.0 for 8 clients. The results are on par with those shown in Table 1. In particular, the performance obtained by the vanilla ResNet architecture on CIFAR10/100 with 4 and 8 clients respectively is: 88.96, 88.6, 56 and 50.6, whereas its performance under the FLHA-distillation baseline is 90, 85.7, 51.1, 44.2. In this example, FLHA-GHN shows success in training with two commonly used non-parametric layers.

## C  HYPERNETWORK INITIALIZATION

As described in Section 4.2 of the main paper, a proper initialization of the hypernetwork weights is instrumental to a successful training of the client networks. In figure 6 this is shown on an example convolutional layer with dimensions: $3 \times 3 \times 64 \times 128$. In both plots, the blue colored histogram shows the distribution of the desired Kaiming weight initialization He et al. (2015b). When the weights are generated by a hypernetwork, a standard initialization of the hypernetwork would generate the orange histogram shown on the left. We show what that histogram looks like after our initialization scheme on the right.

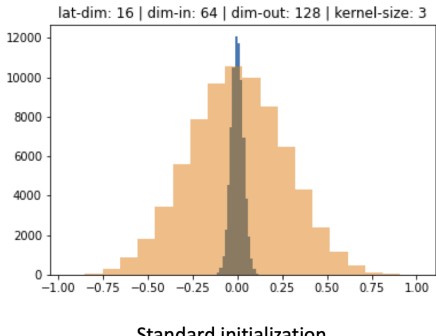 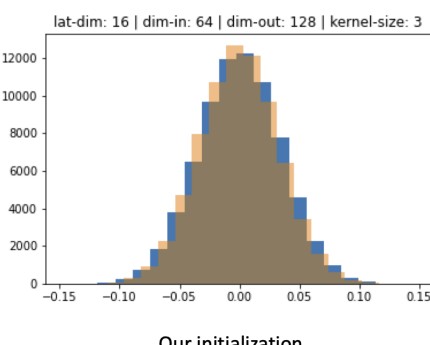

Standard initialization                                        Our initialization

Figure 6: Hypernetwork weight initialization. We compare (blue:) a direct Kaiming weight initialization He et al. (2015b) of a convolutional layer with (orange:) the resulting weight initialization by the hypernetwork, without (left) and with (right) our initialization scheme.

## D  HYPERARAMETER SEARCH FOR OUR FLHA-GHN

We ran an extensive hyperparameter search using Biewald (2020) for a 4 architecture setup using a fixed number of 500 epochs , with 3 different GNN types: GraphConv(Morris et al., 2019), GatedGraphConv(Li et al., 2017), and GraphSAGE(Hamilton et al., 2018); number of GNN layers $T$ from 1 to 8 with latent dimensions between 16 and 128; hypernetwork $H_l$ bottleneck dimension between 16 and 64; learning rates between 1e-4 and 0.1; SGD (with and without cosine scheduler) and Adam(Kingma & Ba, 2014) optimizers with weight decay values between 5e-4 and 5e-6.

## E  IMPLEMENTATION DETAILS: LOCAL DISTILLATION

Baseline distillation from a teacher model trained via standard (same-architecture) FL is done using a distillation loss Hinton et al. (2015): $(1 - \alpha)\text{CE}(y_{pred}, y) + \alpha\text{KL}(y_{pred}, y_{teacher}) \times 2T^2$ where CE and KL denote Cross-Entropy and Kullback Leibler, respectively. The softmax in the KL loss is taken with respect to a temperature $T = 20.0$ and $\alpha = 0.7$. We trained distillation as well as the main FL models for 200 epochs with SGD. The learning rate for training the teacher network (with same-architecture FL) is set to $0.1$ with cosine scheduling. When distilling from the teacher network to the student, we use a learning rate to $0.01$.

## F  IMPLEMENTATION DETAILS: PFEDHN

We found this architecture to be quite sensitive to hyperparameters, hence unlike our method which uses the same set of parameters in all experiments, here we chose the best performing parameters per setting. Hyperparameters sweep on the following parameters: learning rate, number of hidden layers in shared mlp, latent dimension size, optimizer (adam and sgd), and weight decay.

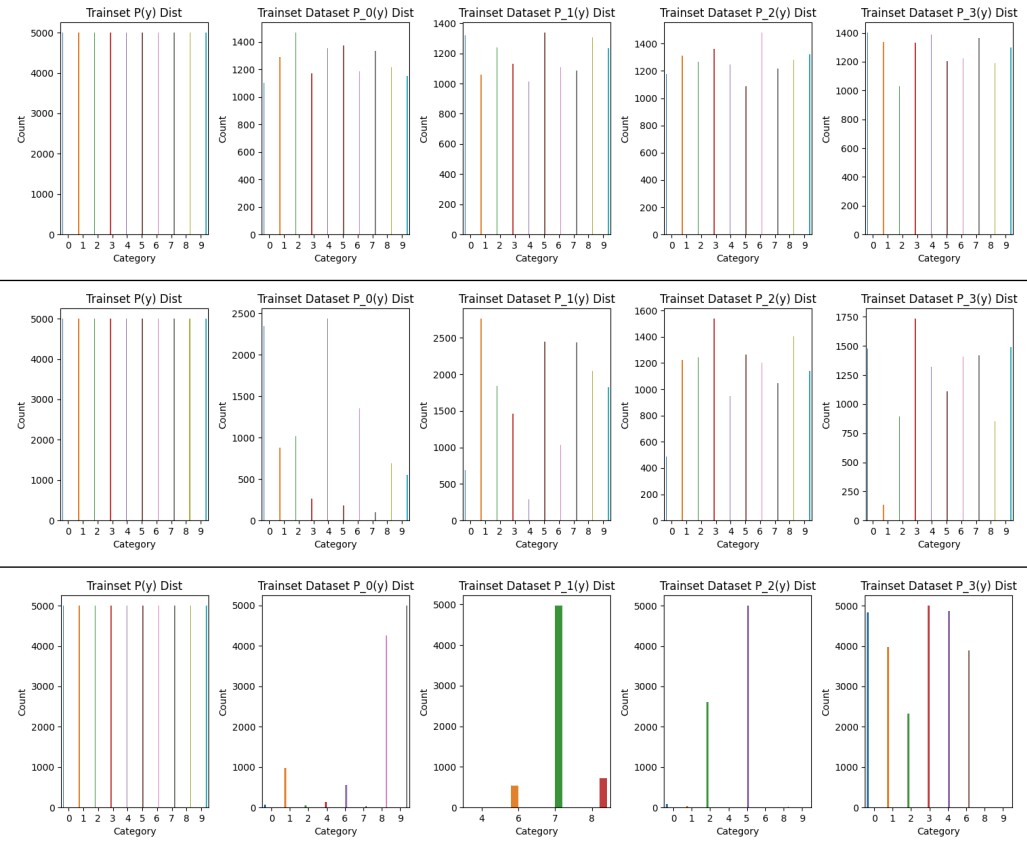

Table 4: Each row in the table shows the unbalanced class distribution for 4 clients, with the original balanced distribution of the left. Three different $\alpha$ values are shown: (top) $\alpha = 100$, (mid) $\alpha = 1$, (bottom) $\alpha = 0.1$

## G  UNBALANCED DISTRIBUTION

In collaborative training between different entities, clients' data may be distributed unevenly. In medical data, for example, this may occur when clinics specialize in certain diseases or use different sensors. Thus, in addition to architectural differences, FLHA can also have data imbalance. Here we study the behavior of FLHA under such unbalanced data distributions.

We follow Yurochkin et al. (2019); Hsu et al. (2019); Lin et al. (2020) and use the (symmetric) Dirichlet distribution, parameterized by a concentration parameter $\alpha$ to split the CIFAR-10 training and test sets between the different clients. Table 4 shows the resulting per-client class distributions. As can be seen, the smaller $alpha$ is, the less balanced the distribution is.

Table 5 shows the performance on CIFAR-10 under 3 different $\alpha$ values: $0.1, 1, 100$. The smaller $\alpha$ is the more unbalanced the distribution is. Class distribution under the different $\alpha$ values are shown in Table 4.

When the training data is unevenly distributed, performance can either be measured in a similar distributed test set or a balanced test set. In the former case, a client would like high performance on local, biased samples. Suppose a hospital specializes in a specific disease and hopes to improve model performance related to that disease through FL. We refer to that a "unbalanced" metric. Alternatively, a client might be interested in balancing its model bias, in which case the performance on the full ("balanced") test set is of interest. We compare our FLHA-GHN against a local training. We also include the usual upper bound performance of standard FL with all clients using the same architecture. Table 5 shows that FLHA-GHN outperforms local training in both "unbalanced" and "balanced" tasks and across all $alpha$ values. Specifically, it can be seen that local training achieves high performance only on test sets of similar distribution, but severely sacrifices performance on sets of balanced

| | | Client 0 | | Client 1 | | Client 2 | | Client 3 | |
|---|---|---|---|---|---|---|---|---|---|
| $\alpha$ | method | unbal. | bal. | unbal. | bal. | unbal. | bal. | unbal. | bal. |
| 100 | FLHA-GHN | 89.7 | 89.5 | 87.6 | 87.6 | 87.0 | 86.2 | 88.3 | 88.1 |
| | Local | 81.8 | 82.4 | 75.1 | 74.2 | 80.5 | 80.5 | 81.1 | 80.3 |
| | Standard FL | 92.9 | 93.7 | 93.2 | 93.7 | 94.1 | 93.7 | 94.5 | 93.7 |
| 1 | FLHA-GHN | 91.1 | 87.1 | 87.4 | 85.5 | 84.9 | 84.6 | 85.0 | 85.5 |
| | Local | 87.8 | 75.8 | 87.3 | 83.5 | 84.4 | 84.0 | 84.4 | 83.5 |
| | Standard FL | 93.8 | 93.0 | 92.9 | 93.0 | 92.6 | 93.0 | 92.9 | 93.0 |
| 0.1 | FLHA-GHN | 94.6 | 62.2 | 98.3 | 28.7 | 92.3 | 34.0 | 92.4 | 61.0 |
| | Local | 93.6 | 51.1 | 96.6 | 27.8 | 90.9 | 25.2 | 90.7 | 54.1 |
| | Standard FL | 69.3 | 53.8 | 58.2 | 53.8 | 42.3 | 53.8 | 49.6 | 53.8 |

Table 5: FLHA with unbalanced distribution. In the table, $\alpha$ corresponds to the level of unbalanced,e.g. $\alpha$=100 (almost uniform), $\alpha$=0.1 (extremelly unbalanced). unbal. and bal. are short for unbalanced and balanced and correspond to the distribution of the test set with unbalanced being the same distribution of each client's training set.

| | GHN Init | From scratch |
|---|---|---|
| Arch 1 (original) | 86.1 | 83.9 |
| Arch 2 (No skip) | 84.2 | 81.3 |
| Arch 3 (Skip first) | 84.3 | 83.7 |
| Arch 4 (Skip last) | 85.6 | 83.6 |

Table 6: Generalization to unseen architectures: leave-one-architecture-out experiment on CIFAR-10. Each row is the accuracy on a held-out architecture while training on the other architectures.

distributions. The same architecture FL also shows a trade-off. Despite being the most performant on the balanced test set, it sacrifices accuracy on local distributions. FLHA-GHN's capability to improve on local training in both tasks can be attributed to the inherent personalization of the network. That is, beyond its ability to adapt to new architectures, FLHA-GHN can also learn a personalized weight prediction according to the client distributions. This result aligns well with the observation of Shamsian et al. (2021).

## H  GENERALIZATION − ADDITIONAL RESULTS

Table 7 shows the performance of the 4 architectures used in our experiments, and how they are influenced when each is replaced by a smaller architecture. The average drop in performance of $2.2 \pm 1.4$ keep the performance well above the local-training alternative.

| Replaced | Arch 1 | Arch 2 | Arch 3 | Arch 4 |
|---|---|---|---|---|
| None | 90.4 | 89.0 | 87.3 | 88.5 |
| Arch 4 | 88.2 | 86.9 | 86.4 | 80.3 |
| Arch 3 | 88.8 | 87.1 | 79.6 | 88.2 |
| Arch 2 | 88.6 | 81.6 | 85.4 | 86.2 |
| Arch 1 | 77.5 | 83.8 | 85.4 | 83.6 |

Table 7: Training with a much smaller architecture shows an average performance drop by $2.2 \pm 1.4$ pts. However, this is well above the local training alternative.

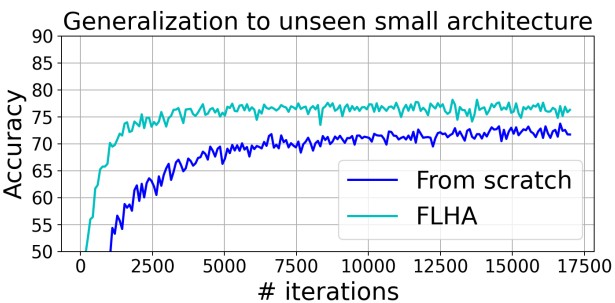

Figure 7: Generalization to a smaller 4 layer CNN architecture. Our method (light blue) quickly ramps up to high performance and maintains a considerable gap compared to training from scratch until convergence(dark blue).

