# OpenReview forum: "Federated Learning with Heterogeneous Architectures using Graph HyperNetworks"
_ICLR.cc/2022/Conference — ICLR 2022 Submitted_

### Official Review · Reviewer_7WPJ · 2021-10-22

**Correctness:** 3
**Technical Novelty And Significance:** 2
**Empirical Novelty And Significance:** 2
**Recommendation:** 3
**Confidence:** 4

**Main Review:**

The proposed method addresses a key challenge in federated learned which is data heterogeneity. The approach is explained in good detail.

The main weaknesses in my opinion:

- The novelty seems limited. The authors use a direct application of existing methods (graph hyper networks).

- The modeling assumptions need to be explained in more detail. It is currently not very clear how the data distributions at clients is related to the neural architectures \mathcal{A}_{c}

- No theoretical analysis of the proposed method (what can you say about the sample size required to train a GHN?).

- The description of the proposed method (Algorithm 1) needs to be expanded. Currently, there seems to be only one single sentence referring to Algorithm 1.

- Numerical experiments use small-scale setups. It is not evident if the proposed method scales to millions of clients.

**Summary Of The Paper:**

The authors study methods for federated learning with clients using different neural network architectures. Graph hyper networks are used to predict useful weights of client-specific neural network architectures.

**Summary Of The Review:**

* The clarity of the presentation needs improvement. First of all, the underlying modeling assumptions on the datasets are not clear. What do you assume about the data distributions $\mathcal{P}_{c}$ and how are they related to the network architectures $\mathcal{A}_{c}$.

* It is nice that the authors write down two research questions at the end of the third paragraph Section 1. However, it is not very clear how and where these questions are answered later on.

* Section 3 is very short. Maybe it could be expanded with a discussion of using graphs to represent neural architectures.

* There needs to be more explanation of Algorithm 1 in the text. How can we tune the input parameter of Algorithm 1 ?  The steps of the inner loop in Algorithm 1 should be fleshed out to provide a sufficient level of detail for implementing Algorithm 1. Should the client architectures \mathcal{A}_{c} be listed as input to Algorithm 1 ?

* The literature review is missing a recent line of work on personalized FL (multi-task learning) using total variation minimization:

D. Hallac, J. Leskovec, and S. Boyd, Network Lasso: Clustering and Optimization in Large Graphs, Proceedings SIGKDD, pages 387-396, 2015.

A. Jung, A. O. Hero, III, A. C. Mara, S. Jahromi, A. Heimowitz and Y. C. Eldar, "Semi-Supervised Learning in Network-Structured Data via Total Variation Minimization," in IEEE Transactions on Signal Processing, vol. 67, no. 24, pp. 6256-6269, 15 Dec.15, 2019, doi: 10.1109/TSP.2019.2953593.

Y. Sarcheshmehpour, Y. Tian, L. Zhang, A. Jung, “Networked Federated Multi-Task Learning,” CoRR abs/2105.12769, 2021

* The following wordings are unclear:
- "... train a strong model ..." what is a strong model ?
- "all clients need to share the same network architecture" pls provide more justification for this claim. in general we can use arbitrary parametric models for clients as long as the parameter vectors are "compatible" e.g. have the same length.

- ".. often run different networks .." what do you mean by "running a network" ?

- "... current FL approaches do not support mixtures of different architectures." im not sure if this is true. consider e.g. using different pre-trained nets as feature extractors for clients and then using the same last layers. as long as we use paramter vectors of same length for clients we can use e.g. network Lasso to implement FL.

- " Since different architectures have different layer compositions, representing layers as nodes allows meaningful knowledge aggregation across architectures. " I do not understand this argument

- "...from some predefined family of models..." what exactly is a family of models?

- "...for a suitable loss function ..." what precisely makes a loss function suitable?

- Figure 2 could include more details e.g. indicate hyper network parameters W_{G} , W_{H} and node representations h_{v}

- "... are learned from data using updates from all clients." unclear

- consistently use abbreviation (e.g. HN) once introduced !

- "... to represent non-parametric operation in the network ..." what is a non-parametric operation ?

- "... see supplementary for more ... "

- " ... and trained in a standard FL."

- " ... when local data is ..."

- "...when local data percentage is at ..."

- "...our GHN can immediately populate it ..."

- " ... we have proposed a new setup ..." pls refer to the relevant equation, algorithm or figure

---

### Official Review · Reviewer_jPp2 · 2021-10-23

**Correctness:** 3
**Technical Novelty And Significance:** 3
**Empirical Novelty And Significance:** 4
**Recommendation:** 6
**Confidence:** 3

**Main Review:**

Strengths:
1.  The proposed solution is novel. The authors adopt a graph neural network with a MLP as a hyper-network to generate model parameters  for each client. This enables each client to protect its own model design as privacy whiling enjoying the benefit of sharing knowledge.
2.  The proposed framework is generalizable to unseen architecture. The authors have shown that when a new client with a different model architecture is encountered the model performs better than learning from scratch.
3.  Experiment results on the x-ray dataset are promising.

Weakness:
1. The variation of selected models of clients is narrow. They are all variants of ResNet with close size.
2. Experiment results on CIFAR-10/100 do not show evident advantage of the proposed method over baseline method.
3. Adding a hyper-network adds extra computation cost. It is suggested to compare training efficiency experimentally.

**Summary Of The Paper:**

This paper studies the problem of sharing knowledge among different organizations without disclosing their model architectures under the framework of federated learning. The authors addressed this problem by sharing a graph hyper-network that can populate model parameters based on different model architectures.

**Summary Of The Review:**

Previous works on federated learning either focus on the mechanism of parameter aggregation or the aspect of privacy. This paper opens a new direction in FL where clients may not be willing to share their unique model designs. From this perspective, I think this paper has promising impact on the research field of FL.

---

> ### Comment · Reviewer_jPp2 · 2021-11-24
> **After reading the author response, my rating is not changed.**
>
> Thanks for trying to addressing my concerns.  As my first concern, the variation of selected models of clients is narrow. They are all variants of ResNet with close size. The authors argued that it is because architectures are assumed to be built from the same "building blocks".  However, I think for models which are not from ResNet family,  they still can share the same "building blocks" with ResNet on a smaller granularity.

---

### Official Review · Reviewer_U48w · 2021-10-26

**Correctness:** 2
**Technical Novelty And Significance:** 3
**Empirical Novelty And Significance:** 3
**Recommendation:** 6
**Confidence:** 3

**Main Review:**

Strengths:
- The paper tackles an interesting and well-needed problem: architecture-agnostic federated learning.
- The paper is clearly organized and written. I can follow it without much effort.
- The method to represent neural architectures as graphs is interesting.

Drawbacks:
- **Generalization experiments seem weak.** For realistic application scenarios, a need is that neural architectures should be chosen corresponding to the computational power (as mentioned in paragraph 2 in the introduction). The networks used in this paper do not differ very much in terms of size (maximally ResNet18), and is thus a little weak to support the motivation. It would be better if you can do the experiments on networks with larger variance in size (e.g. generalize a ResNet152 to a ResNet18 or even smaller).

Further, for figure 7, does this figure imply a <50% zero-shot accuracy of the 4-layer network, and that further adaptation can only be achieved after fine-tuning? Then, does the adaptation result depend on the data upon which to perform fine-tuning? The result does not seem very supportive of the authors' claim of generalization.

- **Ability to deal with very deep networks**. Following the previous drawback, I am considering how the proposed FLHA-GHN can deal with very deep networks, and here is why. Suppose you use a 6-layer GHN, the GHN can distinguish between local graph structures within 6 hops. However, GNNs cannot tell the absolute position of nodes given the same local neighborhoods (You et al. 2019). Thus, for very deep networks (e.g. ResNet152) with highly repetitive structures, networks at different depths but with similar local connection patterns may get very similar weights. I wonder whether the above analysis is correct and would like to see the replies from the authors.

(You et al. 2019) Position-aware Graph Neural Networks. Jiaxuan You et al. ICML 2019.

- **Number of clients**. For the experimental settings, you use 4, 8, 16 clients, while pFedHN uses 50, 100, 500. I wonder what are the reasons behind such a big difference in the number of clients. Any justifications?
- I am curious about how big the graph hypernetwork used in the experiments is. Intuitively, suppose a client wants a network with a linear layer with 16 inputs and 100 outputs. That is 1600 parameters. To generate weights from an MLP, we need an extra linear layer with 1600 output dimensions. Thus, we would need something like $16\times 16\times 100$ parameters for the layer. It is nice if you can show such an analysis.
- **Heterogeneous data**. As shown in Appendix G, the approach to use GHN may lead to worse performance (compared to standard FL) on a balanced test set. This seems a drawback of the proposed method, as FL commonly needs to deal with heterogeneous data. The authors may want to justify the results. Specifically, in practice, we cannot guarantee that the test data distribution at each client is the same as its training data.

Minor issues.
- I fail to understand Table 7.

**Summary Of The Paper:**

This paper proposes a method to tackle the problem of architecture-agnostic federated learning. The primary motivation of this paper is that in some cases of federated learning, different participants may need to use different neural architectures and may not want the exact architecture to be known. To this end, the authors propose a method based on graph hypernetworks. The main insight of this method is that neural architectures can be represented as graphs and their weights can be outputs of a larger GNN. Empirically, the proposed method compares favorably against state-of-the-arts.

**Summary Of The Review:**

The paper has its merits. The paper tackles an important problem. The proposed method is novel and evaluated extensively. The writing is clear and easy to follow. However, the authors may want to justify some minor issues raised in the reviews. I am willing to raise my recommendation of this paper upon further clarifications.

---

> ### Comment · Reviewer_U48w · 2021-11-24
> **After reading the author response, my rating is not changed.**
>
> I am grateful to the authors for trying to address my raised questions. However, the following questions are not sufficiently addressed.
> - **Generalization seems weak.** The concern remains. Specifically, in practice, I cannot see the reason for participants to use different architectures with similar computing requirements. I think it only makes sense when some users are rich in computation resources while others are quite limited, and thus require the experiments to use architectures with very different sizes. The authors fail to clarify the point.
>
> - **Number of clients** As mentioned in previous work [1], the number of clients plays an important role in evaluating FL. Thus, I think the low number of clients makes the experiments slightly unconvincing, despite a fair comparison.
>
>     [1] Fan Lai et al. Oort: Efficient Federated Learning via Guided Participant Selection. In OSDI 2021.
>
> Thus, I cannot be more supportive of the paper than a weak accept.

---

### Official Review · Reviewer_Tk9o · 2021-11-02

**Correctness:** 2
**Technical Novelty And Significance:** 2
**Empirical Novelty And Significance:** 3
**Recommendation:** 3
**Confidence:** 4

**Main Review:**

Strength

1. This paper leverages the GHN techniques to solve the problem of system heterogeneity under FL settings.

2. .The trained GHN model can be generalized to unseen architectures.

3. Extensive evaluation on three real datasets demonstrate the superior performance of the proposed approach for tackling the heterogeneity issue in federated learning.

Weakness

1. The performance highly depends on the quality of trained GHN, and GHN is often affected by the distribution of client data. When the datasets on local devices are highly non-iid, the performance of this method might be significantly dropped.

2. The dimension of the final output is fixed, even though it can be solved by concatenating or slicing the final output. It reduces the flexibility of the generated model. For example, if the output size is 64, then the dimensionality of each layer should be the multiplier of 64. Copying layers and concatenating them together to form a bigger layer is too brutal and may not be a good choice. In practice, the parameter of one layer is not mirrored.

3. In the experiments, what does 'without graph' mean? The accuracy of the proposed method is similar to the 'without graph.'

4. The authors fail to provide the convergence analysis for federated learning, especially the convergence of GHN under FL settings.

5. The paper is more like a straightforward application of the Graph Hyper Network from the centralized training to the federate learning.

**Summary Of The Paper:**

This paper aims to train a GHN model that can predict weights of each layer. The architecture is represented in adjacent matrix format, where a node represents one layer and an edge indicates whether two nodes are connected in given architecture. Each type of layer is represented with a one-hot vector as their node attribute. The outputs are predicted weights of the corresponding layer. Finally, a task-specific object function is minimized by updating the GHN.

**Summary Of The Review:**

Overall, the well designed structure makes the workﬂow clear and easy to follow, but the further analysis and discussion are expected to clarify the contributions in the techniques as well as in the evaluation section.

---

> ### Comment · Reviewer_Tk9o · 2021-11-30
> **Post-rebuttal Update**
>
> Thanks for the authors' efforts in addressing the raised concerns. I have read the author response and other reviews and keep my score due to the following reasons. W4 and W5 have not been addressed. For W3, i.e., Graph importance (“without graph”), as shown in Table 1 in the submission, the accuracy of the proposed method with GHN is very close to and even worse than the 'without graph.' in many experiments. It seems that the GHN or graph structure makes little contribution to the performance.

---

### Author Response · Authors · 2021-11-22
**Authors response**

**General comment**\
The reviewers appreciated the need for a FL method capable of being applied to heterogeneous architecture setups, and the suggestion of using a GHN as a potential solution. We thank the reviewers for providing us with excellent comments for improving the manuscript. We will include these improvements in a future version. \
*The following are the main issues raised by the reviewers:*

**Non-iid data clarifications.** (a) Appendix G shows the experience on non-iid data where four clients train on unbalanced data distributions. "Standard FL" here corresponds to what we call "upper bound" in other experiments, where all clients use the same architecture. We didn't use that term here as the unbalanced iid further endows the GHN with personalization qualities that result in better performance. (b) Our model does not explicitly model the distribution of client data. It is a general problem studied in FL, not specific to GHN, and solutions from pFL can be incorporated into our method. In particular, the Hypernetwork can be conditioned on the client id so that even two clients with the same architecture would have different weights.

**Limited flexibility in architecture.** We assume that architectures are built from the same "building blocks". These could be as rich as needed, but sharing can only occur within the same blocks. Regarding the reviewer's suggestion to “copy and concatenate" smaller computational blocks to construct larger ones, another possibility might be to use our nonparametric layers to construct a graph describing the formation of larger blocks from smaller ones. This way when predicting weights for the concatenated blocks, the HN will be made aware of their structure instead of duplicating the weights.

**Graph importance (“without graph”).** We realized our latest submission was missing a textual description and apologize for the oversight. This row illustrates a version of our method that does not utilize the graph structure. In detail, we propose a simpler way of performing FL with heterogeneous architectures by assigning a HN for every layer type. However it does not leverage the graph to encode the layers’ location within the computational graph.  We can see that except for the occasional large damage to Arch2, under the selected 4 architectures, the graph largely does not provide additional useful information. Based on our choice of architectures, in which a layer type is strongly tied to its location within the graph, the graph encoding is somewhat redundant. However, this is not true in general, as explained in the graph importance section and demonstrated in Table 3.

**A larger variety of architectures** (such as Resnet18 vs Resenet152) will require a greater engineering effort, which we will leave for the future.

**“Zero-shot” generalization.** Compared to training from scratch, generalization to unseen architectures was characterized by quick convergence and improved overall performance when refined on local data. However, performance without refinement varied according to architecture as shown in figures 3 and 7.

**Ability to deal with very deep networks.** The reviewer pointed out a concern regarding the GNN “n-hop” limitation. We address this in our method by implementing an asynchronous graph propagation that follows the order of the computational graph and flows from the top of the graph to the end irrespective of its size.

**Number of clients:** Our choice of client quantity was not based on any special reason other than the manual effort of designing and implementing hundreds of architectures that would work well.

**GGN size:** to populate a layer with P parameters, our MLP requires a layer of size HxP, where H is the hidden layer dimensionality. The more this layer is shared between clients and within the architecture, the more efficient it is.

---

### Decision · Program_Chairs · 2022-01-20

**Decision:**

Reject

**Comment:**

Meta Review of Federated Learning with Heterogeneous Architectures using Graph HyperNetworks

This work investigates a method for federated learning in a neural architecture-agnostic setting. They do this by using a graph hypernetwork to predict the weights of given neural network architectures (which is not exactly known at the onset). The authors conduct federated learning experiments to demonstrate good performance on several real datasets, and also showed that the trained GHN model can generalize (somewhat) to unseen architectures (which are mainly in the ResNet family). Personally, as AC, I find the results very promising, and the experiments show that GHNs are highly applicable to real world applications. But the reviewers outline several weaknesses in the discussion that makes it difficult to recommend acceptance of this paper for ICLR 2022.

The main weaknesses of the work are that application is mainly focused on a narrow family of ResNet architectures (can it be shown to go beyond this? If not, can the writing be improved to show that this is useful enough for many applications?) Reviewer U48w suggested improvements to the generalization experiments, and other details that can be addressed in the writing. Reviewer Tk9o mentioned that this work can be seen as a straightforward application of GHNs (limited novelty), while other reviewers do acknowledge the novelty of the work. I recommend improving the writing to clearly address this and defend why this is not a straightforward application of previous work. With these improvements, I'm confident that this work will be accepted at a future ML conference or journal.

Even though I cannot recommend acceptance, both myself and other reviewers are looking forward to seeing improved versions of this work for publication in the future. As jPp2 also noted, “Previous works on federated learning either focus on the mechanism of parameter aggregation or the aspect of privacy. This paper opens a new direction in FL where clients may not be willing to share their unique model designs. From this perspective, I think this paper has promising impact on the research field of FL.” Good luck!